# “Let’s Use This Mess to Our Advantage”: Calls to Action to Optimize School Nutrition Program beyond the Pandemic

**DOI:** 10.3390/ijerph19137650

**Published:** 2022-06-22

**Authors:** Beth N. Katz, Jessica Soldavini, Kiran Grover, Stephanie Jilcott Pitts, Stephanie L. Martin, Linden Thayer, Alice S. Ammerman, Hannah G. Lane

**Affiliations:** 1Food Insight Group, Berkeley, CA 94707, USA; beth@foodinsightgroup.com (B.N.K.); linden@foodinsightgroup.com (L.T.); 2Center for Health Promotion and Disease Prevention, Gillings School of Global Public Health, University of North Carolina Chapel Hill, Chapel Hill, NC 27599, USA; jessica6@live.unc.edu; 3Department of Nutrition, Gillings School of Global Public Health, University of North Carolina Chapel Hill, Chapel Hill, NC 27599, USA; slmartin@unc.edu (S.L.M.); alice_ammerman@unc.edu (A.S.A.); 4Icahn School of Medicine at Mount Sinai, New York, NY 10029, USA; kgrover895@gmail.com; 5Department of Public Health, East Carolina University, Greenville, NC 27858, USA; jilcotts@ecu.edu; 6Department of Population Health Sciences, Duke University School of Medicine, Durham, NC 27708, USA

**Keywords:** school meal programs, child or adolescent, food insecurity, COVID-19, school wellness, policy implementation science

## Abstract

School nutrition programs mitigate food insecurity and promote healthy eating by offering consistent, nutritious meals to school-aged children in communities across the United States; however, stringent policy guidelines and contextual challenges often limit participation. During COVID-19 school closures, most school nutrition programs remained operational, adapting quickly and innovating to maximize reach. This study describes semi-structured interviews with 23 nutrition directors in North Carolina, which aimed to identify multi-level contextual factors that influenced implementation, as well as ways in which the innovations during COVID-19 could translate to permanent policy and practice change and improve program reach. Interviews were conducted during initial school closures (May–August 2020) and were deductively analyzed using the Social Ecological Model (SEM) and Consolidated Framework for Implementation Research (CFIR). Analysis elicited multiple relevant contextual factors: director characteristics (motivation, leadership style, experience), key implementation stakeholders (internal staff and external partners), inner setting (implementation climate, local leadership engagement, available resources, structural characteristics), and outer setting (state leadership engagement, external policies and incentives). Findings confirm the strength and resilience of program directors and staff, the importance of developing strategies to strengthen external partnerships and emergency preparedness, and strong support from directors for policies offering free meals to all children.

## 1. Introduction

The National School Lunch Program was established in 1946 to achieve two primary goals: (1) to “safeguard the health and well-being of the Nation’s children” and (2) to “encourage the domestic consumption of nutritious agricultural commodities and other food” [1]. The National School Breakfast Program, Summer Food Service Program, and other federal programs were later added to expand nutritious food access for children year-round. This suite of School Nutrition Programs is administered by the United States Department of Agriculture (USDA), and around 30 million children participate annually [2]. The COVID-19 pandemic has highlighted the critical contributions of these programs and potential for innovative change.

School nutrition programs are one of the largest food assistance programs in the United States, but they are not without controversy. School meals are primarily funded through federal reimbursements to school food authorities (SFAs; often school districts) for each meal served. The reimbursement rate is based on student financial eligibility, with some students qualifying for free, some reduced-price, and some full-priced meals [3]. Advocacy around making school meals free for all students (also referred to as “healthy school meals for all”) has grown in recent years. Supporters of healthy school meals for all often cite evidence that school nutrition programs promote better educational outcomes among students of all socioeconomic backgrounds [4], the potential to reduce stigma related to participating in school meals and “lunch shaming” stemming from unpaid meal balances [4,5,6], and the financial stability of maximizing student participation [6]. Concerns about federal policies enabling free school meals for all students have typically centered around the cost to federal and/or state governments and the necessity or appropriateness for students whose families can afford to pay [7,8,9].

The COVID-19 pandemic illustrated the potential for federal policy and local school systems to innovate quickly. The positioning of school meal programs and staff as “essential” became widely accepted during the early months of the pandemic, and policy makers prioritized free and accessible meals for as many families as possible [10]. In March 2020, most governors issued an executive order directing all public schools to close in an effort to limit the spread of COVID-19 [11]. During the emergency school closures, USDA issued a series of waivers that permitted programs to serve meals through the National School Lunch Program—Seamless Summer Option or Summer Food Service Program. This allowed program operators to receive higher reimbursement rates per meal [12]. USDA issued additional waivers allowing administrative flexibility in meal pattern and meal service requirements (e.g., enabling parent pickup, grab-n-go, and delivery options) to facilitate program continuation while mitigating virus spread [13]). Although this was not the first time the USDA granted flexibility in school meal program administration (e.g., during natural disasters), the COVID-19 pandemic was unique in its prolonged, pervasive, and highly visible shock to several interconnected systems that impact school meal programs.

School nutrition directors (SNDs) are trained professionals responsible for administering school nutrition programs at the local SFA level. During COVID-19, SNDs were forced to adapt quickly and continuously to an emerging disaster that exacerbated food insecurity, disrupted supply chains, and presented new and uncertain risks to employees [14]. COVID-19 brought new public attention to the role of school nutrition programs in safeguarding the health and wellbeing of the nation’s children and serving as hubs of resilience in communities [15,16,17]. There is a burgeoning body of qualitative and mixed methods literature describing the critical role of school meal distribution programs in improving food access across the country during COVID-19, the challenges faced and solutions implemented, and the innovative strategies used to ensure that children and families were fed [18,19,20,21,22,23]. In this study, we seek a deeper understanding of the contextual factors that influenced these challenges, solutions, and innovations across multiple levels.

This study describes findings from semi-structured interviews with SNDs across North Carolina during the first six months of the COVID-19 pandemic. We use the Social Ecological Model (SEM) and Consolidated Framework for Implementation Research (CFIR) to describe multi-level contextual factors that influenced program operations, with the goal of identifying factors to address and/or leverage post-pandemic to enable school meal programs to realize their full potential as hubs of resilience in community food systems.

## 2. Materials and Methods

### 2.1. North Carolina Context

In North Carolina (NC) during the 2019–2020 school year, prior to schools closing due to COVID-19, 204 SFAs, including public school districts, charter schools, and residential child care institutions, operated federally assisted school nutrition programs [24]. In fiscal year 2019, the average daily participation in the National School Lunch Program in NC was 835,081 students, [2] approximately 49% of all school-aged children in NC [25]). Of these, 58% received meals for free or at a reduced price [26]. An additional 455,308 breakfasts were served daily [27].

The governor of NC issued an executive order directing all public schools to close for two weeks beginning Monday, 16 March 2020 in an effort to limit the spread of COVID-19 [28]. The NC Department of Public Instruction, the NC Department of Health and Human Services, and the NC State Board of Education were tasked with developing a plan to provide for the health, nutrition, and safety of children while schools were closed. In the first week, more than 1.2 million school meals were served across the state [29]. When school closures were extended, many SFAs usedfederal waivers to continue meal service.

### 2.2. Recruitment

We used several recruitment tactics to recruit school nutrition directors (SNDs) to participate in interviews. First, an email invitation was distributed by a member of the research team via state-wide listserv to SNDs in all public school districts. Second, we sent targeted emails to SNDs in a subset of districts (*n* = 30) selected to maximize demographic diversity and capture regional perspectives. Third, a representative from the state education agency sent a second email invitation via state-wide listserv. The state agency representative also announced the study at a statewide meeting with SNDs in May 2020. A total of 31 SNDs were recruited through these methods; of these, 23 (74%) completed interviews. Interviewees were mailed high-quality aprons for their participation. Study protocols were approved by the Institutional Review Boards of the University of North Carolina at Chapel Hill and the Duke University Health System.

### 2.3. Data Collection

Interviews were conducted by five trained interviewers and recorded via a video conferencing platform between 27 May and 12 August 2020, a period over which USDA’s waivers were issued and re-issued at various time points with uncertain expiration dates [29]. The interview guide was developed iteratively by the research team with input from state and local school nutrition stakeholders. The guide drew from a question bank repository developed by a national working group [30]. We modified the questions from the repository to ensure both local and national relevance and to align with study objectives. The final interview guide is included in Appendix A. At the end of summer 2020 (about 6 months from the start of the pandemic), we determined that we had adequate information power (e.g., sample specificity, based on established theory, narrow study aim, strong dialogue quality, limited comparison in analytic strategy) and thus, further recruitment was not necessary [31].

### 2.4. Data Analysis

Recordings were transcribed verbatim and de-identified. De-identified transcripts were coded and analyzed using Dedoose (Version 9.0.46, Los Angeles, CA, USA: SocioCultural Research Consultants, LLC, 2022). Data analysis was driven by a hybrid deductive/inductive phenomenological approach. We iteratively developed a codebook that combined the SEM and the CFIR. SEM was first conceptualized by Bronfenbrenner in 1979 to describe the multi-level systems of influence on individual behavior change and is frequently employed to guide public health prevention efforts [32]. The CFIR complements the SEM by providing a menu of constructs that describe factors that influence implementation of evidence-based public health efforts across the multiple levels of the SEM [33]. Our overarching framework is illustrated by Figure 1. The framework defines multiple levels through integrating SEM and CFIR terminology, and subcodes defined across each level largely reflect CFIR constructs of most relevance to our study. Transcripts were first divided into excerpts using three time-based codes: Pre COVID-19, During COVID-19, and Beyond COVID-19. Those excerpts were subsequently content-coded using a combination of descriptive and process hierarchical coding across four levels, whereby codes and subcodes were informed by the SEM and the CFIR [34]. All data were double-coded, with a total of five coders meeting in teams of two (one coder left the study team and was replaced by another) to reach consensus. Coders then used matrices for each code to identify potential themes, which were discussed and agreed upon by the full coding team. Pre-COVID and during-COVID codes were condensed to describe factors across multiple levels that influenced the implementation of school meal programs during the pandemic, and the beyond-COVID codes described how those factors may continue to influence implementation beyond the pandemic. Coding methods for all time-based code categories were systematically tracked using meeting minutes, memos, and a detailed audit trail [34,35].

## 3. Results

As described in Table 1, of the 23 SNDs recruited for the study, 21 (91%) represented SFAs that were public school districts and two (9%) represented public charter school systems. SNDs represented SFAs from all eight of the state’s service regions. Based on the National Center for Education Statistics [36], districts were 74% rural, 9% urban, and 17% suburban/town, which is representative of the state as a whole [37]. Interviews averaged 47 min in length (range: 32–84 min).

Below, we present themes elicited from subcodes within each level of our framework in two sections: Implementation Factors Influencing Program Operations During COVID-19, and Beyond COVID-19: Influences on Future Operations. Code definitions and themes are described in more detail in Table 2.

### 3.1. Implementation Factors Influencing Program Operations during COVID-19

*“It may have not have been the prettiest, but we damn sure did it”*. (North Central, Rural)

Although the logistical details of program operations (e.g., how food was procured, prepared, and distributed) varied across our sample, all SNDs described using the waivers to begin serving meals almost immediately after schools closed. Most continued to serve meals in some capacity through the end of the 2019–2020 school year and into summer 2020. Directors described fears of COVID-19 spread, staff safety concerns and labor shortages, food and packaging shortages, and uncertainty around federal policies and reimbursement structures. Key themes related to factors that influenced program implementation are summarized below by SEM/CFIR-based parent codes, and illustrated with additional quotes in Table 2.

“*It wasn’t even a question in our minds. We were going to do something, whether we’ve got funding or whether we didn’t, whatever happened*”. (Western, Rural)

#### 3.1.1. Director Characteristics

We identified several characteristics of the SNDs themselves that influenced program operations: motivation/values, leadership style, and experience. SNDs described being willing to go above and beyond to operate their programs. They worked long hours and played many roles to ensure their programs met their ultimate goal of feeding kids.

*“We could serve masses and masses of people from our serve line. That’s what we do all day long”. We’re experts at that. But when you threw in the, “Oh, by the way, you’re going to take everything on a bus”. We were like, “Okay. Yeah, we could do this. Because we’re—I call ourselves ‘the bendy flexibles’”.* (Northeast, Rural)

##### Motivation and Values

We identified several motivations for SNDS to continue operations during early pandemic days, including their connection with the students and a sense of purpose behind what they do for a living. They were motivated by the knowledge they were feeding children who needed the food to survive and thrive.

*“When people ask me what I do for a living, I said, ‘I feed hungry children. I do that with a smile and a hell of a lot of pride’”.* (Southeast, Rural)

SNDs were also motivated to be good stewards of funding and resources (e.g., by using commodities and items they already had on hand), and keep grab-n-go food appealing to students despite it being pre-packaged. A sense of responsibility for the safety and morale of their staff was commonly mentioned. Directors described specific new approaches to supporting their team members, such as offering new employee incentives (e.g., childcare).

##### Leadership Style

Whereas SNDs were not always involved in decision making at the district level, many did report increased communication with local leaders by necessity. Several SNDs noted that being involved in decision making allowed them to act proactively and plan before the school closures in order to begin meal service immediately. In addition, many directors described the necessity of using a team-based approach. They gathered input from staff and modified operations based on staff recommendations. They also realized the need to publicly recognize the hard work of their staff.

*“I wrote a grant for two purposes, employee incentives and for supplies. So…after our supplies were purchased, I divided up [the remaining funds] by the number of employees, and at my end of the year recognition banquet, I offered them that remaining sum per employee”.* (Southeast, Suburban/Town)

##### Experience

Although most SNDs acknowledged it was difficult to be prepared for the massive changes required for pandemic operations, many described drawing on past experiences (e.g., hurricane-induced school closures) to increase their confidence and guide decision making. Their familiarity with various areas of program management and general need to be thrifty and flexible during financial challenges prepared them to lead during the pandemic.

*“The thing with school nutrition is that we roll with the punches. If you tell us one day that we’ve got to do something different, we’re going to say, ‘Okay, let me figure it out. We got it. Let’s just roll on with it’”.* (North Central, Rural)

#### 3.1.2. Implementation Stakeholders

Both internal staff (e.g., food service staff, administrators, teachers, maintenance, and transportation staff) and external partners (e.g., community organizations and individual volunteers) played a critical role in pandemic operations.

##### Internal Staff

Overall, SNDs described staff as “unsung heroes” who were dedicated, flexible, creative, and willing to take on new roles in order to prepare and distribute food.

*“Those employees have become delivery drivers. Those employees have become whatever you wanted them to be. No one has really complained because we’re all in it for the same purpose, and that is to make sure those kids are being fed. I mean, I have my Spanish interpreter that learned how to drive a big box truck”.* (Piedmont-Triad, Rural)

Many SNDs described staffing challenges due to COVID-19 fears, resentment among working staff toward staff who were being paid while staying home [38], or feeling under-appreciated by community members. Many SNDs worked with districts to fill these gaps (and provide employment opportunities) to non-food-service district employees (e.g., transportation, maintenance, teachers, administrators) who distributed and delivered food. Working with these employees not only helped to mitigate staff shortages, but also improved relationships and communication across departments that had not worked together previously.

*“A lot of times my staff feels like they’re not part of the school, and I think this has really changed that. We’ve had the collaboration of teachers’ assistants, custodians, principals, school resource officers. Principals were helping load buses in the morning with coolers and helping lift things. All different groups that pitched in and helped out because they wanted to make sure the kids got fed. It was a great experience”.* (Western, Suburban/Town)

##### External Partners

Many directors reported strengthening and/or forming new partnerships during the pandemic, including with the county health department, community organizations (e.g., churches), national organizations (e.g., No Kid Hungry), other school districts, and local food suppliers (e.g., restaurants). These organizations often provided “no strings attached” support via grants, packaging and food supplies, volunteers, or funds, which was generally welcomed if they met specific pandemic-related needs.

*“Within 24-h, I had 14 churches show up with all of the supplies that they had had in their fellowship halls…hinge containers, and bags, and full sheets. I just made them a list of ‘this is what I need.’ And [they] went out and recruited it from restaurants. Got it from Sam’s. They made trips and sales to try to fill our need, and they did”.* (Piedmont-Triad, Rural)

A few SNDs diverged from this theme, noting that this type of support may have been random and sporadic, and thus was not a reliable method for keeping programs operating or meeting the needs of hungry students in their community.

#### 3.1.3. Inner Setting

The Inner Setting describes factors associated with the success of programs during school closures in the school, school district, and local community. Named factors included implementation climate within the community, local leadership engagement, available resources, and structural characteristics.

##### Implementation Climate

Even though a few SNDs experienced pushback from program participants or community members that hindered program operations or impacted staff morale, most described an increased appreciation of and support for school meal programs among caregivers, students, and the community. This acknowledgement and recognition increased morale and facilitated smoother operations because people wanted to help and were sympathetic through difficult changes. Importantly, the positive attention was felt in contrast to the limited and sometimes negative attention SNDs felt their programs received prior to the pandemic.

*“…I feel appreciated. When they came in and said feeding the kids is the No. 1 priority, [we] really felt appreciated. We don’t always feel like we’re a priority. We always feel at the other end of the scale. But people have come to realize that we are important”.* (Northeast, Rural)

*“We’ve gotten lots of thank-you’s, thank-you’s, thank-you’s. We’ve had parents actually mail thank-you’s straight to our office. We’ve had kids decorate their sidewalks”.* (Piedmont-Triad, Rural)

##### Local Leadership Engagement

In general, SNDs described school and district leaders (e.g., superintendents) as supportive from the start—either through taking swift action and helping to solve problems, or by being open to input from the SNDs and not “obstructing” their efforts.

*“My superintendent stepped up and offered [incentive] to transportation and school nutrition employees that were serving and each stop got $50 a day in addition to their salary as a bonus pay”.* (Southeast, Suburban/Town)

The urgency of decision making often necessitated more frequent communication between district leadership and SNDs than before the pandemic. Some SNDs noted that district leadership also gave more attention to helping programs communicate with families to keep them aware of site locations, meal pick up or delivery times, menus, and other critical pieces of information.

##### Available Resources

With the influx of demand for packaging supplies, pre-packaged foods, and personal protective equipment across the country, most SNDs experienced supply shortages and skyrocketing costs. Many used USDA commodities and other food they had stored, and the waivers enabled them to problem-solve and use creative new strategies to procure, prepare, and distribute food. However, operations were hindered by supply chain issues, and SNDs often expressed concern about the amount of money that was being spent without certainty of reimbursement, particularly in districts where programs already faced financial challenges.

*“They want us to do all these things, but then they’re like, ‘Well, you have to buy it yourself.’ And I’m like, we’re already losing money. We’re trying not to, but our system has lost money probably for the last five or six years, and we’re trying to reverse it, but it’s a hard trend to reverse, and it’s hard to recoup that money, and so you want me to do this stuff, but I don’t have any funding to do it”.* (North Central, Rural)

At the same time, as the above SND acknowledged, increased revenue from the waivers could enable purchase of items that could be used long-term, such as sealing equipment or delivery vehicles.

*“Why would we buy $30,000.00 cargo vans when this is a one-time thing? Well, who knows if it’s a one-time thing or not? And if we have the funding now for it, we should go ahead and get them because then we can keep our participation up year-round because we can use those cargo vans for 10 years”.* (North Central, Rural)

##### Structural Characteristics

Various community structural characteristics influenced program operations, such as district size and geography. Several SNDs in smaller districts noted there were both advantages and challenges to being smaller. Having fewer children to feed meant that they could sustain operations with a smaller supply. However, with limited storage capacity and purchasing power, it could be difficult to secure needed food and supplies on short notice, necessitating more creative solutions. One SND described partnering with neighboring districts to meet minimum shipping requirements.

*“We’re a small community, we’re about 2100 students, and [nearby County] is also relatively small. When the shortages hit, we were like, let’s call the vendors and see if we can get direct ship straight from the vendor. And a lot of the vendors have minimum shipping amounts. So the problem was, like [nearby County], there’s no way we could handle a minimum 12 cases or 12 pallets of pizza. So I reached out to a director in [nearby county] and said, ‘We may not be able to do it individually, but we could do it together. We could split the inventory.’ So, that’s what we did. We found vendors that had individually wrapped products, like pizzas and sandwiches and all kinds of breakfast items. And we were able to meet the minimum orders”.* (North Central, Rural)

Another advantage of smaller districts was more knowledge of the areas in their community where students would be most in need of meals, and as a result of area eligibility waivers, they could place sites in those areas. The waivers also allowed them to use distribution methods that made the most sense for their area—in some cases, delivery via school buses or vans was preferable to distribution at school or community sites due to distance or road quality.

*“We’re not doing individual meals delivery. We’re going to where we know the kids are. We’re a rural county, so trailer parks, those kinds of things. And those kids, they’re out there every day, even in the rain. Like today, it was pouring down rain. They’ll still be out there, looking for their meals”.* (North Central, Rural)

#### 3.1.4. Outer Setting

Outer Setting factors influence implementation at the outermost level, such as state- and federal-level leadership, statewide networks, and broad policies and practices. We elicited factors across three categories.

##### State Leadership Engagement

SNDs nearly all agreed that school nutrition leaders in the state’s education department were a “steadying force” that facilitated successful program operations. State leaders took their role as intermediary seriously, bridging the divide between federal waivers that were issued and the directors tasked with implementing waivers on the ground. SNDs praised their constant and efficient communication and appreciated their convening all-director calls that enabled them to ask questions, express concerns, and hear about what other SNDs were doing.

*“[State] School Nutrition Program has done a wonderful job of providing us with guidance and assistance as they’ve had it. They’ve [said], ‘We don’t have any answers yet, but this is the guidance we’re giving you.’ They were very good at getting ahold of us any time they actually did decide on something”.* (Northwest, Suburban/Town)

##### External Policies and Incentives—School Nutrition Waivers

All SNDs used the waivers, and most acknowledged that the federal waivers that granted implementation flexibility and higher reimbursement were essential to continuing operations amidst the soaring expenses of labor, goods, and delivery costs and for making sure students in their districts could access meals. As mentioned above, the waivers enabled directors to make decisions that best fit the needs of the families in their communities.

*“I am usually a fairly vocal critic of USDA. But they have 100% been outstanding for pulling out all those barriers through this. I recognize why those provisions are put in place during the normal summer meal program to prevent abuse and make sure that kids are the ones that are getting the food. But I’m so glad that they have given us the freedom to give the parent the meal so they don’t feel like they have to bring their babies to a place where the child would be at risk. That they’ve allowed us to set up in areas that ordinarily you wouldn’t even think of putting a meal area and you wouldn’t have participation. But then, you put it in there and you have 80 kids come out. I mean, that just shows me how high the need is across America right now. And so, USDA has absolutely stepped up…”* (Southwest, Rural)

However, many directors noted that the ways in which the waivers were handed down (e.g., last-minute changes, shifting expiration dates) resulted in unnecessary contingency planning, costs, emotional distress for directors, and confusion for families. There was constant concern waivers would expire and not be extended, which would cause a scramble for resources and a drop in program participation, leading to a drop in revenue. Many SNDs reported worrying this would lead to staff layoffs, including school bus drivers and other departmental staff who had been working with school nutrition programs to deliver meals, and would reintroduce the known challenges of the program (e.g., stigma) while failing to account for the additional challenges brought on by the pandemic.

*“[Expiring waivers are] going to really financially impact your programs, and you’re probably going to put some people out of work, because I really don’t think we can sustain what we’re doing if we drop off our paying kids”.* (Piedmont-Triad, Rural)

One SND noted that to prepare for future disasters, the federal government should develop a disaster plan that streamlines the waiver process to ensure smooth program operations.

*“As responsive as USDA was, I think it probably would be smart if they had a disaster plan in place already. That would flip the switch and activate six waivers at once instead of having to meet and vote and then get [a new one] every week. That’s the part that’s been a little bit chaotic is every week, something new comes down and you’re like, ‘I hope I got it right.’ But I can tell from the amount of questions on our weekly calls with [the state agency] that there’s still a lot of confusion”.* (Southwest, Rural)

##### External Policies and Incentives—Other Policies

Other federal policies also influenced meal program operations during COVID-19: Pandemic Electronic Benefits Transfer (P-EBT), a program that provided families the cash equivalent of school meal funding, and funds allocated to school nutrition programs by the state government through the Coronavirus Aid, Relief, and Economic Security (CARES) Act. School food authorities needed to follow specific guidelines in using these funds [39].

When probed about P-EBT, SNDs were generally supportive of the policy but critical of the rollout. SNDs were asked to provide data about free and reduced-price meal program participants for P-EBT eligibility, and many noted that families often asked them about it even though they had very little knowledge about P-EBT because it was managed by a separate state agency. Several SNDs suggested that their programs were more appropriate recipients of federal funding than sending funds directly to families. Additionally, some SNDs perceived or observed a dip in participation in their programs in the days following P-EBT distribution.

*“I’m so glad that P-EBT exists… but it has been such a nightmare, logistically. There have been so many mistakes along the way. There’s been people who didn’t receive cards, people who have received cards but it was in a deceased person’s name…And school nutrition has very little that we can do about any of these problems”.* (Western, City)

When SNDs discussed CARES Act funding, they had similar sentiments as with the waivers—they appreciated the assistance but either had limited guidance on how to use funds or were hesitant to spend money on items that they were concerned would not be reimbursed as indicated.

### 3.2. Beyond COVID-19: Influences on Future Operations

Although most SNDs struggled to envision the future of their programs (likely because the interviews occurred during a period of tremendous uncertainty and day-to-day changes), many did suggest that their programs would benefit from continuation of many COVID-era policy changes.

Importantly, many SNDs felt that pandemic operations had demonstrated that a permanent policy change to make meals free for all students could reduce programs’ administrative burden and “microscopic management,” increase their operating budgets, and enable SNDs to focus on meal quality, food, and nutrition education, cultivating a skilled school nutrition workforce, and expanding partnerships in the community. Several directors also noted that the stigma associated with receiving school meals for free or at a reduced price would be resolved if meals were free for everyone.

*“I think the federal level just needs to suck it up and say every child gets to eat. The paid kids should not have to hold that program up. It’s not fair to them in my opinion because, if their parents pay or not, it’s not the kid’s fault. So, I think federally the program should just be straight across the board universal. Come up with a plan. You just did it for 12 weeks”.* (Northeast, Rural)

*“You don’t pay for school. You don’t pay to ride the bus. Why are you paying for meals? I think 110% I would be in favor of [free school meals for all]. I’m not saying it comes without problems. Certainly, it does. …I just don’t see why you should have to pay for school meals. Just fund your districts properly. I’m talking about really from the federal level. And then also from the state level because our state does not really fund [school meals]. We get $0.30 for every reduced child at breakfast, which is great. But a little district, that might be $2000. That’s not a lot of money”.* (Southwest, Rural)

In addition to funding districts to provide free meals for all, some SNDs also expressed the need for additional operational funding. While SNDs have had to be thrifty and flexible for years, and these attributes benefited their programs during COVID-19, they also benefitted from the additional funds provided to pay for needed resources, staff, and high-quality food.

*“…We need funding. Not just during an emergency, but throughout the school year…when we’re feeding our children and our counties, we should not be struggling to fund our program. But now, all this money has come up to help us do what we’re doing. We need it during the school year, too, because a lot of the [programs] are struggling. A lot of people don’t know that we get no funding except for our reimbursement on our free/reduced meals. And that barely covers just the meals and the labor. But then you’ve got employee benefits to pay… If we are state-employed, we should be getting state funding for those benefits, not coming out of the school nutrition budget, where we could do more for our children, and offer more, higher-quality products. It’s a shame you have to look when you’re doing your menu, what can we afford, when we should be offering our best”.* (Southwest, Rural)

## 4. Discussion

Our analysis of operations during COVID-19 shed new light on policy and practice efforts that could improve school nutrition programs beyond the pandemic. We constructed themes that reflected the innovation and resilience of school meal programs while in crisis mode. During the early months of the pandemic, when health and economic stability were threatened nationwide and food insecurity became a top concern, school food programs and their operators moved front and center [40,41,42,43]. Our study highlights the extent to which program directors in a Southeastern US state innovated, adapted, and stretched to prioritize the children they feed. We also identified contextual factors related to director characteristics, other stakeholders, and the inner and outer setting that influenced program operators’ ability to adapt and innovate during the pandemic.

SNDs were motivated to operate during the pandemic to continue meeting the needs of children in their school districts, to keep programs financially solvent, and to keep staff safe and employed. Their dedication, creativity, and leadership in the face of constantly evolving challenges reinforces findings across other qualitative studies [18,19,20,21,22,23]. We also identified the influence of directors’ existing relationships and prior experience with disasters-induced school closures. Flexibility and resilience were key themes in this and other recent studies [18,21,22]. The SNDs in our study were “bendy flexibles” who were able to “roll with it” despite all the challenges faced in procuring and distributing school meals.

Federal, state, and local actors and policy decisions often facilitated further SND flexibility; however, SNDs sometimes felt that these same entities put up roadblocks and/or failed to act in timely and maximally supportive ways. Using directors’ own words, we issue clear calls to action for policy and systems changes to continue to enable tailored innovations that strengthen program operations. Our findings corroborate other pandemic-era studies in other areas of the U.S. that reported on the importance of internal and external partnerships [19,20,21,22], and underscore the value of cultivating collaboration for community food security regardless of individual state or regional policy climate. In our study, new internal partnerships with other school district departments and personnel not only helped maintain operations, but also made staff feel more integrated into their districts. Thus, the strengthening of internal partnerships is an important innovation to carry forward to streamline operations and legitimize school nutrition. Steps should be taken to ensure that districts continue to align school nutrition efforts with other internal departments and to provide the necessary resources to do so. Further research should investigate the extent to which internal and external partnerships are maintained and formalized post-pandemic, identify strategies for sustaining them, and test their impact on local food systems more broadly.

More investigation is needed into the role of structural characteristics, particularly district size and geography, in hindering or enabling school meal operations. SNDs described various challenges during pandemic operations because of district size, but they also described the ways in which their size or locale facilitated creative solutions. A 2020 study used geospatial analysis to examine meal site placement in urban areas during COVID-19 and found more sites located in higher poverty, higher minority areas [44]. This study, as well as studies conducted prior to the pandemic in the summer months (when sites are placed in communities, not just schools), have primarily focused on urban areas [44,45,46,47,48]. Future research could integrate qualitative implementation data with mapping data [48] to investigate local factors that influence reach and implementation by rurality, district size, and other relevant geographic indicators, particularly as supply chain issues have persisted throughout the pandemic [49]. This could inform how preparation and delivery models could be tailored to meet local needs, both during emergency and non-emergency operations.

We expect future public health and climate-related disasters will require more frequent pivots to emergency operations. SNDs adapted swiftly in the early months of COVID-19, and steps should be taken to ensure more preparedness for future disasters. Our findings endorse suggestions from Patton et al. to develop training manuals, attend to employee safety concerns, increase speed of communication, use social media, and promote the role of school nutrition employees as “essential” for future preparedness [18]. Additional recommendations based on our findings include having resources readily available (e.g., maintaining storage infrastructure, allowing districts to retain more than the current limit of three months’ operating expenses), giving SNDs a seat with local leaders at the decision-making table, and maintaining contact information of local partners and other nearby school districts. Future preparedness also requires federal policy leniency and a budget for emergency operations [18,19,23], as USDA’s traditional reimbursement formula is, as described by Kenney et al., “untenable” during school closures [23].

SNDs in our study were nearly united in stating the potential positive influence of federal legislation to shift from the current model of having free, reduced, and paid meal categories to “healthy school meals for all,” regardless of household income. Informants in other qualitative empirical studies conducted during COVID-19, which have been conducted in states with varying political climates, have also endorsed healthy school meals for all, [21,23] as have various national advocacy organizations [50,51,52]. The United States Congress is tasked with Child Nutrition Reauthorization (CNR) every five years, an opportunity to improve regulations and systems that govern school nutrition programs. These policies have not been meaningfully updated since 2010. Opponents of healthy school meals for all have cited concerns around costs [7,8,9], the need to collect household income data for education funding [53], concerns about eroding meal nutrition standards [50,51,52], or have simply argued that the funding for the programs is adequate [51,52]. SNDs from North Carolina, as well as other states across the country [18,21,23], do not agree that funding is adequate, as many have felt that they have had to be too thrifty, sacrificing program quality. SNDs felt that healthy school meals for all could eliminate stigma and that reducing the administrative burden of collecting family income paperwork could free up staff time for meal preparation and funds to procure high-quality food locally. Our data sends a uniform message to federal policy makers from SNDs. As one of our SNDs stated, “We have asked for years for this program to be funded to feed all kids. If it’s ever going to happen, it needs to happen now”.

Finally, this and other studies document the heightened awareness and appreciation for school meal programs from other district personnel, community members, and the media during the early pandemic [18,19,20,21,22,23]. SNDs point to a potential culture shift around school nutrition, but the question remains how to keep the support and acknowledgement of value going as post-crisis systems resume. As part of this call to action, researchers, including our team, should prioritize better dissemination of research findings to non-research stakeholders (e.g., writing op-eds or creating policy briefs to share with congressional committees) to elevate SNDs’ experiences and translate those experiences to policy and practice change.

Our findings should be interpreted with consideration to several limitations. SNDs have highly demanding jobs during the best of times, and this was exacerbated during the COVID-19 pandemic. Even so, 23 SNDs participated in interviews at a time when stress and uncertainty were at an all-time high. These circumstances may have led us to a sample of directors particularly motivated to share their experiences and perspectives; thus, it is unclear whether our sample is representative of all North Carolina SNDs. Additionally, the perspectives of SNDs do not necessarily reflect the experiences of other school nutrition staff or students and families who received school meals. Future studies should investigate staff, student, and family perceptions of school meal programs during and beyond the pandemic to better inform policy decisions at all levels. Finally, although we included directors from diverse North Carolina regions, our sample does not allow us to explore differences by district size, geography, or length of the directors’ tenure. Some directors indicated important differences in how these factors influenced pandemic-related experiences and program operations, which warrants further exploration.

## 5. Conclusions

School nutrition programs faced numerous operational challenges during the early months of the COVID-19 pandemic yet continued serving meals via innovative methods and partnerships as a result of USDA waivers. Through this study, we identified multi-level factors that influenced the success of these innovations, including characteristics of the SNDs themselves, characteristics of internal and external stakeholders, the district and community (inner setting), and the state and policy climate (outer setting). As stakeholders like USDA, state and federal policy makers, and program operators contemplate expanding school nutrition programs based on COVID-related innovations, they should be mindful of the multi-level factors that our SNDs identified as crucial to ensuring that programs realize their full potential as hubs of resilience within community food systems.

## Figures and Tables

**Figure 1 ijerph-19-07650-f001:**
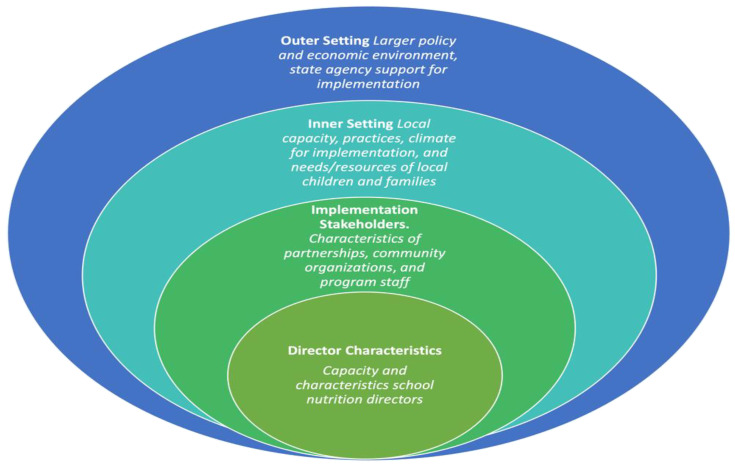
Social Ecological Model (SEM) and Consolidated Framework for Implementation Research (CFIR)-based coding framework of multi-level implementation factors influencing school meal program operations during COVID-19.

**Table 1 ijerph-19-07650-t001:** Characteristics of districts and charters represented by participating school nutrition directors (*n* = 23).

Characteristics	*n* (%)
*Sponsor Type*	
School District	21 (91.3%)
Charter School	2 (8.7%)
*Locale Classification*	
Rural Remote	2 (8.7%)
Rural Distant	8 (34.8%)
Rural Fringe	7 (30.4%)
Suburb/Town	4 (17.4%)
City	2 (8.7%)
*Region*	
Southwest	3 (13.0%)
North Central	6 (26.1%)
Northwest	2 (8.7%)
Northeast	4 (17.4%)
Southeast	2 (8.7%)
Western	3 (13.0%)
Piedmont-Triad	2 (8.7%)
South Central	1 (4.3%)

**Table 2 ijerph-19-07650-t002:** Themes and representative quotes related to implementation factors influencing school meal program operations during the early months of COVID-19, including future anticipated influences.

Code Definition	Themes	Representative Quote(s)
**Director characteristics:** capacity and characteristics of the school nutrition director (interview participant) that may have influenced implementation during COVID-19		
Motivation/Values: Motivations and prioritization driving directors’ decisions	Directors were driven by personal connections with students and knowledge that they are improving student outcomes Directors’ decisions had to balance meeting students’ needs with feasibility and keeping programs financially solvent During COVID-19, directors were also motivated by a responsibility to keep their staff safe and maintain morale	We have children’s health and nutrition interests at the heart of what we do. It’s not about just putting some porridge on their plate and sending them on their way. It really is setting them up with the fuels they need for their body to grow and their brains to grow and to learn (Southwest, Rural) My second priority is making sure my staff is safe. We preached distance, spread out your work stations, supplied masks and gloves. Probably the scariest part of this was just making sure we kept our people healthy and safe, and nobody come down with this. And thankfully, none of our employees did. It makes me feel like we did our job. (Southwest, Rural)
Leadership: Directors’ leadership style/skills (e.g., management style, efforts to build staff moral and skills, role in decision-making, communication)	COVID-19 operations required increased communication with state/district leaders and with their staff across levels Directors supported/empowered their staff in new ways during COVID-19 (e.g., soliciting input, making decisions as a team, and concerted efforts to boost staff skills, safety and morale) Directors worked long hours and took on additional roles than they typically held	I wrote a grant for two purposes, employee incentive and for supplies. So …after our supplies were purchased, I divided up [the remaining funds] by the number of employees, and at my end of the year recognition banquet, I offered them that remaining sum per employee. (Southeast, Suburban/Town) We still really try to send home homestyle meals during COVID. That’s why we were trying to do that hot meal, because I think hot meals are important to kids. We still really focused on trying to send home our fresh fruits and vegetables, and that was one thing I challenged my employees to do. I said, ‘Okay, let’s think out of the box. Let’s think of what we have.’ (Piedmont-Triad, Rural) We don’t know what our job is anymore. I’ve been the warehouse person. I’ve been the off-loader. I’ve been the mopper, the sweeper, the dish washer. I’ve done it all. On top of having to plan the food, to talk to the vendors, be at these meetings. They’re like, ‘Miss [name], you need to go home. It’s 8:30. Why are you in the office still?’ I said, ‘Because when y’all leave at 2:00, I get to do my job. From 7:00 to 2:00, I’m your person. From 2:00 to 8:00 or 9:00, I’m doing the director job.’ (Northeast, Rural)
Experience: Directors’ skills related to food service and prior years of experience/past jobs, and confidence in their ability to adapt during COVID-19	Programs have always had to be thrifty and “roll with the punches,” and many directors had prior experiences adapting during natural disasters Directors relied on their program management expertise to make confident operational and financial decisions during COVID-19.	Our motto is ‘school nutrition, we roll with it and do the best we can with the hand we’ve been dealt.’ And again, I’m proud of what my team has done to feed our children. (Southwest, Rural) Procurement is important to me, so I make a point of knowing the manufacturing rep. I look at different companies to bring in. I’ve got a large warehouse. I look at every dollar we spent as my children’s dollar, so I make sure everybody in my organization, from the server, to the person who delivers the food, to my superintendent, knows that every dollar we’re going to get the bang for the buck. (Southeast, Rural)
**Implementation Stakeholders:** Characteristics of partnerships, community organizations, staff for implementation		
Internal Staff: Internal food service staff and school district staff, including school/district administrators, maintenance staff, teachers, transportation department	School food staff were “unsung heroes” who were highly motivated, adapted their usual schedules and job duties to continue feeding students Staffing challenges included: being short-staffed, staff with their own childcare needs, resentment toward staff using leave policies, low morale, and having to fund additional needed staff Personal connections to students played a significant role in building staff morale Pandemic meal operations necessitated new comradery across district departments and enabled meal service staff to develop new skills	These folks have really put it all out there, running bus routes every day. The movement of food has been outstanding. They are just moving cases and mountains, and we’re doing it all by school bus at 45 miles an hour. (North Central, Rural) I think some of my staff members feel maybe a little bit resentful that they are some of the only staff members in the district who’ve been required to be in the public since March. I think there’s just a general feeling of under-appreciation for what’s being done. People really are taking a personal risk by being out there and distributing the meals. (Western, City) It has been unreal how people have pulled together. Bus drivers want to work to be able to carry the meals. Teacher assistants, bus monitors have wanted to work to continue to be paid. It is unreal, the people that have just worked together and wanted to come together to work. Student Services, social workers, nurses… have been instrumental. (Northeast, Rural)
External Partnerships: non-school district partners, community organizations, and volunteers	Relationships with other districts, local and national organizations, county health department, local food suppliers were strengthened out of necessity Many community organizations and citizens provided “no strings attached” support via grants, supplies, volunteers, or funds. Generally this was welcomed in order to get needed supplies like PPE, packaging, new equipment, but was not viewed as a reliable or consistent resource	We have worked very, very closely with the health department, a lot closer than we have been in the past. They’ve always been a very close community partner with us, but I think it’s been even closer now because we’ve had to have that close communication. (North Central, Rural) We set up a unique situation. We created a contract between [district] Public Schools—we were the sponsor of the summer program. And then, [partner organization] was our vendor that we purchased meals from. Then, [district] Public Schools Foundation was a partner with us who did the accountability piece—that actually did the meal counts and the meal service and things. And me being alone, I had no way to be at all these sites at one time. So, it worked out perfectly that the three groups worked together (North Central, City) We’ve had a lot of folks reach out and wanna help, and we leaned on them to do that. Had a lot of donations in the mail, which I didn’t have that much first eight months on the job. But after COVID hit, people would just send us $50, $100, to help feed kids. They didn’t really ask for anything in return, just wanted the money to go to feed kids. (Northwest, Rural)
**Inner Setting:** Supply, capacity, local practices, local climate for implementation, and needs/resources of local children and families		
Implementation Climate. Existing local climate (e.g., how program is rewarded and supported) and capacity to carry out programs	Attention given to school meals during COVID-19 helped alleviate the “bad rep” and prior under-appreciation of the program On occasion, community or family pushback was demoralizing	I actually think that one of the best things that I hope comes out of it, is that people will see, understand, and hopefully appreciate, that school nutrition does a mighty fine job, particularly when you put into the equation the size budget and the Federal Guidelines that we have, right? I mean, it’s nobody has the same guidelines as school nutrition anywhere. And then to put your meal out on your budget of a couple of bucks, I mean, it’s a joke, really, I mean, nowhere else do you find the same set of circumstances, but who’s there every day, those ladies, we’re there every day. And I think certainly initially there’s gonna be more appreciation for that. How long-lasting that will be, who knows? But we’ll take it how long that we can get. I don’t think people will be quite so scathing in moving forward about free lunch, not ‘only poor kids eat free lunch’. (Northeast, Rural) There were some [parents] that would call and say, ‘My kid’s really picky. Can I just get this?’ Or, ‘Can you post the menu?’ And I’m sitting there thinking, ‘Yeah, no.’ We just told them, ‘This is what we have. If you sign up and we’re delivering, we’re gonna deliver to you every day. If it’s something that one kid doesn’t like and another kid does, you can share. If you know somebody that can utilize it if it’s something that you’re not gonna utilize, then hand it off to somebody else.’ (Northwest, Suburban/Town)
Local Leadership Engagement. Experience with district leadership (e.g., commitment, involvement and accountability) with implementation	Operations got up and running due to either swift actions or “non-obstructive” support from district leaders Communication channels between leaders, directors, and families were more accessible than prior to the pandemic	The [school] board here supported us. They wrote letters, they reached out to the senators saying, this was a need because we do have food insecure students here in [county] and a large number of them. They were willing to write those letters and anything else that I needed them to do…so, they’ve been very supportive. One of our board members even used her church vans to help us distribute meals out to students. She would drive out to trailer parks and park their van there and make sure the students had meals up until we could get the yellow buses running. (Piedmont-Triad, Rural) In the first couple months, it was very important that the different departments were all communicating with one another, and everybody was on the same page and for the different departments to have a level of empathy and understanding towards the other departments. (Northeast, Rural) Our school system blasted it out over all their social media streams…most of the programs that would help community also knew about it, so I feel like the information was there on how to get meals. I noticed that there was a lot of Facebook chatter from the local county sites. I feel like the message was out for people to be able to locate us. (Southwest, Rural)
Available Resources and infrastructure to procure, prepare and distribute food.	With many programs in need of the same products (e.g., packaging supplies, certain types of foods, personal protective), shortages were common and costs skyrocketed Programs benefited from existing resources such as larger food stores or budget surpluses The waivers enabled strategies such as school buses and vans that expanded programs’ capacity to reach students	We had to completely redo the menu, we went from what we call batch cooking to basically individually-wrapped curbside service to-go things. When COVID first hit, everybody was after the same product. There wasn’t creativity. It was, everybody’s got to have this particular product. So, the shortages were really hard at first. (North Central, Rural) We have very much been trying to be good stewards over what we have already purchased and use our inventory. God forbid we let something expire and then have to throw it in the trash. We’re very mindful of that. (Southwest, Rural) We started feeding 17 March, and at that time didn’t have any of the school buses. So that first weekend, we weren’t hitting 4000 students. Then when we incorporated the yellow school buses, we went up to 8000 per day. (Piedmont-Triad, Rural)
Structural Characteristics. Contextual characteristics of communities that influence implementation	District size and geography influenced program operations including procurement and distribution (e.g., smaller districts had less supply on hand or purchasing power but also knew the needs of community better) Adaptations were made to better reach students in their community (e.g., by moving sites) and keep staff safer as new data emerged	I feel like [school nutrition] programs in smaller districts are not nearly supported as those of us that are in larger districts that have a larger tax base. I don’t know if it’s additional reimbursement or what [is needed], but it’s a lot harder in a small district to react to things like this. (Southwest, Rural) The staff was just incredible. We never missed a lick. They understood the importance. We’re probably a little bit different mindset than say [bigger County]. We see this every day and we understand that these meals that these kids get at school are probably about the most nutritional, and for some, the only [meals they get] So, there’s a battle here that we fight constantly, not even including COVID. (North Central, Rural) It wasn’t feasible for us to go door-to-door like a lot of districts had done mainly because of where these buses would have to stop. Since we have like three or four main highways that run through the whole county and a lot of these children are off these main roads, it was a safety issue. (North Central, Rural)
**Outer Setting:** Larger policy and economic environment, and state agency support for implementation		
State Leadership Engagement. Commitment, involvement and accountability of leaders and managers	State agency leaders were seen as a “steadying force” that increased directors’ readiness and capacity to operate programs Regular communication with state agencies through webinars/calls created a “community of practice” among directors	Talking to [state agency] every week, they keep us informed of what’s going on at the state and federal level. They’re the “attaboy” crew. They let us voice concerns and they answer a lot of repetitive questions. Because even though we don’t talk individually, we’re all facing the same challenges. From people I know in other states, North Carolina has stepped up. The state agencies down to our nutrition directors, have done everything to ensure we’re still feeding kids. (Western, Rural) The directors across the state that are willing to share what they’re doing and their plans, and being willing to help each other is a huge benefit for our profession. We’re all in similar situations, but not really the same when it comes to the makeup of our district and how things are handled in each county. Being able to bounce ideas off each other and see what other folks are doing, and hearing their plans is very helpful when trying to create something from scratch. (Western, Suburban/Town)
External Policies and Incentives: Waivers how waivers facilitated or hindered program operation	Initial USDA waivers broke down barriers to reaching families and enabled programs to focus on their bottom line of feeding kids Frequent waivers expirations and changes prevented directors from planning ahead (e.g., making new purchases) and/or led to financial risks Waivers were described as an experiment for how “healthy school meals could work, especially in the summer months	We’ve never had access to those places before, the trailer parks, low-income housing, because we don’t have any way to transport our food. So, this is not a new issue. It’s just exacerbated because of the situation. It’s just really more glaring now, when we don’t have the ability to get to those kids. I mean, there are kids. We wanna feed them. We just can’t get to them. (North Central, Rural) We’ve had to make some tough financial decisions but, again, we haven’t gotten the state support to help offset those costs [packaging, transportation, staffing]. (Southeast, Rural) I worry that we won’t have as many students participating…and I will probably end up being overstaffed. That’s one of the reasons that I really hope that they will continue to allow the summer feeding program to be functioning after August, because that will help a lot, in that respect… I’ve spoken with my administration, and explained, or shared my fears with them of not having the participation that we need to support our staff, and just to let them know that that’s definitely a possibility that this the school year is going, could very well be a really bad budget year for us. The revenue’s not going to be there like it has been, and we were fortunate that with the number of students we were serving March through May was a lot higher, and we actually made more during that time than we do in a typical school year, that that has provided a little bit of extra padding for us revenue wise. But that will most definitely be depleted this school year. Any gains we made will be gone, and we may have to kick in local funds to help keep our folks employed, keep everybody working. (Western, Suburban/Town)
External Policies and Incentives: Other Policies: how other policies (e.g., pandemic EBT, CARES act) facilitated or hindered program operation	Feelings about the usefulness of pandemic EBT and financial resources (e.g., CARES Act funding) were mixed, but most SNDs were critical of the rollout	But there’s so much stipulation to that CARES funding that was given too that it was, “=’Yes. You can use it for this. Whoa, wait a minute. We’re not sure that you can use it for that. Hold on a second. Let’s read a little bit more into this. Oh no, you can use it for that now’. So, then we thought we were going to lose it. So, that was a little tricky… Larger districts need it for different things than what smaller districts do. So, I just personally think that USDA needs to do a better job at that. (North Central, Rural) We found out we could use PRC125 to pay people outside of school nutrition that were dealing with food service. Now that helped us tremendously. (Northeast, Rural) And there have been waivers requested to help districts financially through this time. The money we normally would’ve received through reimbursement, the government then took that money and divided it up and they’re giving the families P-EBT cards. So, pandemic EBT cards to help the families. So, they’re saying that this money would’ve been spent on feeding children, so they’re gonna let the parents feed the children with that money. The downside is that money would’ve been paying our staff. It would’ve been buying our food supplies and our equipment. So, I understand it helps the families, but now you’ve left the districts with big losses (North Central, City)
**Beyond COVID-19**		
Programs would benefit from the continued presence of COVID-era factors across levels: additional funding, more autonomy, new partners, and positive media and community attention. Pandemic operations shone light on the feasibility of universal free meals, and should be a catalyst to increase political will for funding and resources		I guess the big word here would be, you’ve got to be flexible. You’ve got to be flexible during this time—any kind of emergency. And willing to learn new things, do things different than you ever have before in any emergency. And do the best you can. Stay positive as much as you can through it because there were times, I’ll be honest with you, I didn’t even like myself at the beginning of all this because it’s stressful. It is very stressful. And you’ve got the weight of feeding these children on your shoulders. And you’ve got to figure out how to do it overnight, really. (Southwest, Rural) I have been saying this for years, and so have my colleagues not only here in North Carolina, but across the United States. I think we need to do away with pricing at all for our students. It should be universal free meals for all of our children. USDA needs to quit using school nutrition as a weapon. We feed hungry kids. Give us the resources and the support that we need to do [it]. And it needs to be a priority, not an afterthought. We don’t need to fit in after everything else has been figured. We need to be part of the beginning of the conversation, not at the end of it. (Southeast, Rural) I think my thought process has been strengthened as a result of the pandemic, but I felt this for many years that I’ve worked in child nutrition, that it needs to be a universally free program. And I believe, without having done any research, I believe that the money spent on the microscopic management and follow-up for the federally funded programs, all the administrative money, if that was just invested back into the program, there wouldn’t be a problem in terms of affording it. […] Now would be the perfect time to really go for a push on that, because it’s all in upheaval anyway. It’s all a mess. So, let’s use the mess to our advantage to bring in some of the change more quickly than we might’ve been able to otherwise. We have to make sure we do it in a better way. We can’t go back to what we were doing before. There’s just no way that that should be right for anybody to think that that is an acceptable situation. (Northeast, Rural)

## Data Availability

Portions of de-identified data from qualitative datasets may be made available upon request.

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
