# Peer review of "“Let’s Use This Mess to Our Advantage”: Calls to Action to Optimize School Nutrition Program beyond the Pandemic"

_ijerph, 2022, doi:10.3390/ijerph19137650_

Round 1
Reviewer 1 Report
This paper provides very timely, important, and detailed analysis of offering school meals to students in need during the pandemic through the school nutrition programs.
Specific comments:
1. Page 3, line 102 – “ … 835,081 students (approximately 49% of all school-aged children in NC), …”
Do authors have any information related to the rate of eligibilities for all school-aged children in NC?
2. Page 4, lines 158-159 – Coding methods were systematically tracked using meeting minutes, memos, and a detailed audit trail.
What do authors mean by “using the meeting minutes, memos, and a detailed audit trail?” Is the information from those ways of collection used for pre-COVID and during-COVID factors or just for the implementation beyond the pandemic? Please clarify.
3. Page 4, Table 1—Charter school, 21 (91.3%) under Sponsor type.
This number does not match the description in line 167, “… and two (9%) represented public charter school systems.” Please double check.
4. Page 5, Table 1 – Rurual Remote, 2 (8.7%) under Locale Classification.
There seems to a duplicated “Rural Remote” listing on the Table 1. Please double check.
5. Page 12, line 215– SNDs are had was a sense of responsibility for the safety and morale of their staff.
Please modify the sentence to be “SNDs had a sense of responsibility for the safety and morale of their staff.
6. Page 12, line 218—“.. pay, childcare)..”
There is an extra period “.” at the end of the sentence.
Author Response
Point 1: This paper provides very timely, important, and detailed analysis of offering school meals to students in need during the pandemic through the school nutrition programs.
Thank you for this comment.
Point 2: Page 3, line 102 – “ … 835,081 students (approximately 49% of all school-aged children in NC), …” Do authors have any information related to the rate of eligibilities for all school-aged children in NC?
We added the following sentence, Pg 3, Line 104-105: “Of these, 58% received meals free or at a reduced price [30]”
Point 3: Page 4, lines 158-159 – Coding methods were systematically tracked using meeting minutes, memos, and a detailed audit trail. What do authors mean by “using the meeting minutes, memos, and a detailed audit trail?” Is the information from those ways of collection used for pre-COVID and during-COVID factors or just for the implementation beyond the pandemic? Please clarify.
We revised sentence Page 4, Lines 168-169 to say “Coding methods for all time-based codes were systematically tracked using meeting minutes, memos, and a detailed audit trail”
Point 4: Page 4, Table 1—Charter school, 21 (91.3%) under Sponsor type. This number does not match the description in line 167, “… and two (9%) represented public charter school systems.” Please double check.
Thank you for catching this, we have corrected the accidental duplication in the table.
Point 5: Page 5, Table 1 – Rurual Remote, 2 (8.7%) under Locale Classification. There seems to a duplicated “Rural Remote” listing on the Table 1. Please double check.
Thank you for catching this, we have removed this duplicate row.
Point 6: Page 12, line 215– SNDs are had was a sense of responsibility for the safety and morale of their staff. Please modify the sentence to be “SNDs had a sense of responsibility for the safety and morale of their staff.
Thank you, we have corrected this typo.
Point 7: Page 12, line 218—“.. pay, childcare)..There is an extra period “.” at the end of the sentence.
Thank you, we have corrected this typo.
Reviewer 2 Report
This study aims to analyze endeavors to maintain school nutrition programs during the difficult conditions of the COVID-19 pandemic in the US, by using the Social Ecological Model and the Consolidated Framework for Implementation Research. Using data obtained from long interviews with school nutrition directors, this study presents many interesting and valuable points not only from the viewpoint of academic research, but also from the viewpoint of practical policy implementation. As pointed out in the last paragraph of the Discussion section, one of the limitations of this study is that the perspectives of school nutrition directors do not necessarily reflect those of students and families. It is undeniable that the directors may have actively exaggerated endeavors and outcomes of their activities, in order to sustain and strengthen the school nutrition program, and it is unclear to what extent the results of the analysis are affected by bias. However, despite the limitation mentioned above, this paper certainly provides useful insights regarding what we need to keep in mind while maintaining efforts to prevent children's nutritional intake from deteriorating, in the context of unanticipated social turmoil. Therefore, I believe that this paper is worth publishing.
Author Response
Point 1: This study aims to analyze endeavors to maintain school nutrition programs during the difficult conditions of the COVID-19 pandemic in the US, by using the Social Ecological Model and the Consolidated Framework for Implementation Research. Using data obtained from long interviews with school nutrition directors, this study presents many interesting and valuable points not only from the viewpoint of academic research, but also from the viewpoint of practical policy implementation. As pointed out in the last paragraph of the Discussion section, one of the limitations of this study is that the perspectives of school nutrition directors do not necessarily reflect those of students and families. It is undeniable that the directors may have actively exaggerated endeavors and outcomes of their activities, in order to sustain and strengthen the school nutrition program, and it is unclear to what extent the results of the analysis are affected by bias. However, despite the limitation mentioned above, this paper certainly provides useful insights regarding what we need to keep in mind while maintaining efforts to prevent children's nutritional intake from deteriorating, in the context of unanticipated social turmoil. Therefore, I believe that this paper is worth publishing.
Response 1: Thank you for this comment. We have elaborated in the limitations section to further emphasize the potential for bias in our analysis as a result of our sample, Page 20, Line 611.
Reviewer 3 Report
The topic is interesting and addressed in a different new way.
The manuscript is well organized, clear and straight to the point.
The methodology is explicitly explained. BUT it would be better to introduce in details the frameworks used (SEM & CFIR), and how they were combined to create the framework (Fig 1) of this study.
Results are very well presented especially the table linking all codes to themes and supported by participants' quotes.
Limitations, strengths and future recommendations are clearly identified.
References are good and updated.
Author Response
Point 1: The topic is interesting and addressed in a different new way.
The manuscript is well organized, clear and straight to the point.
Results are very well presented especially the table linking all codes to themes and supported by participants' quotes.
Limitations, strengths and future recommendations are clearly identified.
References are good and updated.
Response: Thank you for these comments.
Point 2: The methodology is explicitly explained. BUT it would be better to introduce in details the frameworks used (SEM & CFIR), and how they were combined to create the framework (Fig 1) of this study.
Response: We have expanded on our description, Page 4 Lines 154-157: The framework defines multiple levels through integrating SEM and CFIR terminology, and subcodes defined across each level largely reflect the CFIR constructs of most relevance to our study. We also added details to introduce the results, Page 4, Line 181: Below, we present themes elicited from subcodes within each level of our framework in two sections…
Reviewer 4 Report
First, this is a well-written piece of qualitative work. Easy to read and an interesting update of the experience of implementing school food programs during the pandemic. Good work!
A few very minor points for your consideration.
Higher level: Given the nature of the special issue-- can you bring the focus a bit more to "innovative" practices that emerged during the pandemic and what implication these might have moving forward in a non-pandemic environment? This ties into the final conclusion where you state that the SNDs "innovated quickly"--basically, it makes me want to understand the kinds of ways they pivoted, what influenced how they pivoted and then the implications for the future...which is what you have basically outlined here. The amount of information, however, can feel a bit overwhelming and probably just needs an overall tightening up of the summaries of your findings.
Introduction:
- line 85-86, can you make it clear that the objective is to look at the contextual factors DURING covid that influenced....; also, is manuscript the correct word to use here? Once it is published, it will not longer be a manuscript, yes?
- you introduce waivers in the findings and I found that part hard to follow because you did not introduce waivers and how they work in the introduction-- but regardless of whether you introduce it in the introduction, you need to provide more context about the waiver system to make that part of the findings more clear.
METHODS:
- consider adding a portion that describes how you knew that you had collected "enough" data (ie. saturation)
RESULTS:
- Table 1 is not formatted well ("sponsor type" and "local classification" probably needs to be in a different font or something like that?
-Line 175- 178: formatting issues re indents and being so close to the bottom of the table
TABLE 2: no need for this in the results section-- it is too much-- this would be better as an appendix or a supplemental file, I think.
- Overall generall formatting in the findings was a bit erratic (sub-headings were not consistent, the quotes were in their own paragraphs-- can you center them and turn them into blocks when they are more than two lines long? Or, when they are less than two lines long, can you incorporate/embed some of the quotes right into the sentences so they aren't their own paragraphs?
- overall language when referring to the findings-- you use phrases like" "SNDs expressed several motivations...." (line 207)-- I would do an edit and use more tentative language when referring to what you found. Did all the SNDs express this? Or did a majority of them express this? Or did "some" of them express this? Surely the experience of 23 different participants were not entirely universal? Doing this helps the reader to understand the magnitude/strength of the specific findings. It is okay if not everyone said the same thing-- or even if just a few people said the same thing--it does not mean there isn't "truth" there. I would revise this throughout the findings section as it occurs frequently where it sounds like ALL of the SNDs always said the same thing and in were 100 percent agreement.
- a few minor spelling errors--a good close copy edit should do the trick here. (eg. line 249, line 566, etc)
DISCUSSION:
-You bring in some quotes into the discussion- from my perspective, this is you introducing other findings in the discussion which is not sound practice. either incorporate these quotes into the findings or remove these parts where it seems new results are being introduced
- You have some really rich findings here about this complex implementation environment-- while this is excellent, it can feel overwhelming for the reader. Can you package up the summary in the conclusion a bit better/tighter? For example, you say that the SNDs "...innovated quickly to continue serving meals"... what do you mean by this? Can you summarise what you mean by this? (related to one of my first comments at the top).
Overall, nice work!
Author Response
Point 1: First, this is a well-written piece of qualitative work. Easy to read and an interesting update of the experience of implementing school food programs during the pandemic. Good work!
Response 1: Thank you for this comment
Point 2: Higher level: Given the nature of the special issue-- can you bring the focus a bit more to "innovative" practices that emerged during the pandemic and what implication these might have moving forward in a non-pandemic environment? This ties into the final conclusion where you state that the SNDs "innovated quickly"--basically, it makes me want to understand the kinds of ways they pivoted, what influenced how they pivoted and then the implications for the future...which is what you have basically outlined here. The amount of information, however, can feel a bit overwhelming and probably just needs an overall tightening up of the summaries of your findings.
Response: We added more explicit statements throughout the Discussion to tighten up this point. Page 18, Line 511, 518-519, page 19, Line 533-534, 540, 539, 550, 568, page 20, Line 591. We defer to the editor to streamline if all are needed. We also re-wrote our Conclusion to better reflect the issue theme, Page 20-21, Lines 622-632.
Point 3: Line 85-86, can you make it clear that the objective is to look at the contextual factors DURING covid that influenced....; also, is manuscript the correct word to use here? Once it is published, it will not longer be a manuscript, yes?
We have added “during COVID” to Page 2, Line 86 for clarification, and changed “manuscript” to “study” on line 88.
Point 4: You introduce waivers in the findings and I found that part hard to follow because you did not introduce waivers and how they work in the introduction-- but regardless of whether you introduce it in the introduction, you need to provide more context about the waiver system to make that part of the findings more clear.
Response: We added additional information on the waivers on Page 2, Lines 67-72: USDA issues a series of waivers that permitted programs to serve meals through the National School Lunch Program – Seamless Summer Option or Summer Food Service Program, which allowed program operators to receive higher reimbursement rates per meal [16]. USDA also issued additional waivers allowing administrative flexibility in meal pattern and meal service requirements (e.g., enabling parent pickup, facilitating grab-n-go or delivery options), to facilitate program continuation while mitigating virus spread ([17].
Point 5: Consider adding a portion that describes how you knew that you had collected "enough" data (ie. saturation)
Response: Given the nature of the research question and our study population’s busy schedules, our goal was not saturation; rather, we sought to recruit a specific sample and have quality data (e.g., information power) within our recruitment resources. We added the following to Page 3, Lines 138-142: At the end of summer 2020 (about 6 months from the start of the pandemic), we determined that we had adequate information power (e.g., sample specificity, based on established theory, narrow study aim, strong dialogue quality, limited comparison in analytic strategy) and thus, further recruitment was not necessary [35].
Point 6: Table 1 is not formatted well ("sponsor type" and "local classification" probably needs to be in a different font or something like that?
Response: We have italicized the section headings within the table, but defer to the formatting team to verify this change.
Point 7: Line 175- 178: formatting issues re indents and being so close to the bottom of the table
We have moved that paragraph above Table 1, now page 4, Lines 181-184.
Point 8: TABLE 2: no need for this in the results section-- it is too much-- this would be better as an appendix or a supplemental file, I think.
Response: Given that Reviewer 3 noted the utility of this table for reviewing the results, we left it in. However, we will defer to the editorial team to decide whether it should be re-submitted as a supplemental file.
Point 9: Overall generall formatting in the findings was a bit erratic (sub-headings were not consistent, the quotes were in their own paragraphs-- can you center them and turn them into blocks when they are more than two lines long? Or, when they are less than two lines long, can you incorporate/embed some of the quotes right into the sentences so they aren't their own paragraphs?
Response: The quotes and headers were formatted this way in the post-submission process, but we agree that they are erratic and have adjusted them. We defer to the copy editing team to verify this change.
Point 10: overall language when referring to the findings-- you use phrases like" "SNDs expressed several motivations...." (line 207)-- I would do an edit and use more tentative language when referring to what you found. Did all the SNDs express this? Or did a majority of them express this? Or did "some" of them express this? Surely the experience of 23 different participants were not entirely universal? Doing this helps the reader to understand the magnitude/strength of the specific findings. It is okay if not everyone said the same thing-- or even if just a few people said the same thing--it does not mean there isn't "truth" there. I would revise this throughout the findings section as it occurs frequently where it sounds like ALL of the SNDs always said the same thing and in were 100 percent agreement.
Response: We have revised the language throughout the results section to be more tentative, with the exception of instances where most responses were the same (e.g., staff working hard) E.g., Page 12, Line 210, Line 220, pg 13, Line 230, 235, 246-247, 269, pg 15, Line 326-327, 333, 337, 355, Page 16, line 387, 399, 415, pg 17, Line 420, 447, 459, 465, 467, pg 18, Line 470, 492
Point 11: a few minor spelling errors--a good close copy edit should do the trick here. (eg. line 249, line 566, etc)
Response: We have corrected these spelling errors and re-reviewed the full manuscript to correct additional typos, and hope that the copy editing team can identify and correct any additional errors.
Point 12: You bring in some quotes into the discussion- from my perspective, this is you introducing other findings in the discussion which is not sound practice. either incorporate these quotes into the findings or remove these parts where it seems new results are being introduced.
Response: We appreciate this perspective. We debated this as well, but included them in the discussion to drive home several of our key points. The quote now on lines 593-594 further reinforces the theme about healthy school meals for all; thus, we decided to leave that. The following quote does not reinforce a theme from our data, but emphasizes the importance of sharing findings beyond academic journals in one of our participant’s own words. We have removed this quote for this revision, but defer to the editorial team to make the final decision:
“What y’all [the study team] are doing is great, getting all this collective data… but you know what? The people that already do it, get this information. The pastors and superintendents and the general public, they don’t go to these same meetings and they don’t go to these same workshops…spend some of these grants on billboards, and commercials, and radio time, I mean, don’t sit down and write papers and…nice little grants. People need to understand. Put billboard signs up, put radio time out, put TV commercials out…don’t show us a PowerPoint at the National School Conference because we see it, we know it, we created the data. It’s the people out here that we got to get involved.” (North Central, Rural)
Point 13: You have some really rich findings here about this complex implementation environment-- while this is excellent, it can feel overwhelming for the reader. Can you package up the summary in the conclusion a bit better/tighter? For example, you say that the SNDs "...innovated quickly to continue serving meals"... what do you mean by this? Can you summarise what you mean by this? (related to one of my first comments at the top).
Response: We re-wrote our Conclusion to better reflect the issue theme, Page 20-21, Lines 624-632.